# Treatment success rate and associated factors among drug susceptible tuberculosis individuals in St. Kizito Hospital, Matany, Napak district, Karamoja region. A retrospective study

Ronald Opito[1], Keneth Kwenya[2], Saadick Mugerwa Ssentongo[2]*, Mark Kizito[3], Susan Alwedo[2], Baker Bakashaba[2], Yunus Miya[2], Lameck Bukenya[2], Eddy Okwir[4], Lilian Angwech Onega[2], Andrew Kazibwe[2], Emmanuel Othieno[5], Fred Kirya[6], Peter Olupot Olupot[7,8]

1 Department of Public Health, School of Health Sciences, Soroti University, Soroti, Uganda, 2 Directorate of Program Management and Capacity Development, The AIDS Support Organization, Kampala, Uganda, 3 Department of Internal Medicine, School of Health Sciences, Soroti University, Soroti, Uganda, 4 Directorate of Programs and Community Services, Baylor College of Health Sciences, Kampala, Uganda, 5 Department of Pathology, School of Health Sciences, Soroti University, Soroti, Uganda, 6 Department of Anatomy, School of Health Sciences, Soroti University, Soroti, Uganda, 7 Department of Public Health, Faculty of Health Sciences, Busitema University, Mbale, Uganda, 8 Mbale Clinical Research Institute, Mbale, Uganda

* ssentongomugerwasaadick@gmail.com

## Abstract

### Background

Tuberculosis (TB) is the leading cause of death among infectious agents globally. An estimated 10 million people are newly diagnosed and 1.5 million die of the disease annually. Uganda is among the 30 high TB-burdenedd countries, with Karamoja having a significant contribution of the disease incidence in the country. Control of the disease in Karamoja is complex because a majority of the at-risk population remain mobile; partly because of the nomadic lifestyle. This study, therefore, aimed at describing the factors associated with drug-susceptible TB treatment success rate (TSR) in the Karamoja region.

### Methods

This was a retrospective study on case notes of all individuals diagnosed with and treated for drug-susceptible TB at St. Kizito Hospital Matany, Napak district, Karamoja from 1st Jan 2020 to 31st December 2021. Data were abstracted using a customised data abstraction tool. Data analyses were done using Stata statistical software, version 15.0. Chi-square test was conducted to compare treatment success rates between years 2020 and 2021, while Modified Poisson regression analysis was performed at multivariable level to determine the factors associated with treatment success.

**Data Availability Statement:** All relevant data are within the manuscript and its Supporting Information files.

**Funding:** This publication was partially (data collection) supported through SUNRIF, Soroti University research and innovation fund, Round1, Award No SUNRIF 2022/22 to Ronald Opito. The views and opinions of the authors expressed herein do not necessarily state or reflect those of the funder. There was no additional external funding received for this study.

**Competing interests:** The authors have declared that no competing interests exist

**Abbreviations:** AIDS, Acquired Immune Deficiency Syndrome; aRR, Adjusted risk ratio; CDC, United States Centers for Disease Control and Prevention; DLG, District Local Government; EPTB, Extrapulmonary Tuberculosis; HC, Health Center; HIV, Human Immuno-deficiency Virus; LAM, Lipoarabinomannan; MoH, Ministry of Health; MUAC, Mid upper arm circumference; PBC, Pulmonary Bacteriologically Confirmed; PCD, Pulmonary Clinically Diagnosed; PEPFAR, President's Emergency Plan for AIDS Relief; REC, Research Ethics Committee; TB, Tuberculosis; TSR, Treatment Success Rate.

## Results

We studied records of 1234 participants whose median age was 31 (IQR: 13–49) years. Children below 15 years of age accounted for 26.2% (n = 323). The overall treatment success rate for the study period was 79.3%(95%CI; 77.0%-81.5%), with a statistically significant variation in 2020 and 2021, 75.4% (422/560) vs 82.4% (557/674) respectively, ($P$ = 0.002). The commonest reported treatment outcome was treatment completion at 52%(n = 647) and death was at 10.4% (n = 129). Older age, undernutrition (Red MUAC), and HIV-positive status were significantly associated with lower treatment success: aPR = 0.87(95% CI; 0.80–0.94), aPR = 0.91 (95%CI; 0.85–0.98) and aPR = 0.88 (95%CI; 0.78–0.98); respectively. Patients who were enrolled in 2021 had a high prevalence of treatment success compared to those enrolled in 2020, aPR = 1.09 (95%CI; 1.03–1.16).

## Conclusion

TB TSR in Matany Hospital was suboptimal. Older age, poor nutrition, and being HIV-positive were negative predictors of treatment success. We propose integrating nutrition and HIV care into TB programming to improve treatment success.

## Introduction

Globally, tuberculosis (TB) is the leading cause of death among infectious disease. It is estimated that 10 million people fell ill with TB in 2020, albeit only 5.8 million were notified, due to COVID-19 ramifications [1]. In 2020 the global TB treatment success rate (TSR) was 86% and 1.5 million people died of the disease. Notable co-infection was HIV which was an underlying cause of mortality in TB in 14% [1]. The African region contributes 25% of the global TB burden and has a TSR of 79% which is still below the End TB target of 90% [2]. There is significant variation in TSR across sub-Saharan Africa (SSA), with a gradual decline over the years [3, 4]. In uganda, TSR has improved from 72% in 2019 to 78% in 2020, however this is still below 95% as recommended by world health organization (WHO) [5].

Uganda is among the 30 high TB/HIV burden countries [6]. The national TB statitics remain poor. For instance, Uganda's TB incidence rate of 230-330/100,000, prevalence of 253/100,000 and HIV prevalence among those aged 15–64 of 6.2% are among the worst disease indicators reported [7–9]. A large proportion, 41% of patients with TB in Uganda have HIV coinfection [7–9]. Treatment outcome is one of the key indicators of TB program measure of the quality of TB care [8, 10]. A number of factors have been identified to affect TB TSR in many settings, including being male, older patients, previously trated for TB [11, 12], HIV co-onfection [12, 13], and drug stock-outs [14].

The Uganda Ministry of Health national TB and Leprosy program and partners have recommended evidence-based interventions to reduce TB incidence and improve TB treatment outcome [15], but data on its progress remain scanty.

In 2019, the Karamoja region in Northeastern Uganda had one of the highest TB burdens in the country [16]. To date information on factors associated with TB TSR across Karamoja is limited partly because the inhabitants of the region are mainly normadic pastoralists. Within the region, the Ministry of Health and other researchers approximate that only 50% of the TB patients complete treatment successfully [9, 16–18]. This is below the national average of 78% in 2020 [5].

Examining factors associated with the low TSR in this region is important for designing tailor-made interventions that may be used to address the barriers to achieving a high treatment success rate. It is very important that people who have TB disease are treated successfully to eliminate pools of infection sources and interrupt disease transmission. Therefore, the orverarching aim of this study was to determine the drug susceptible TB treatment success rate and the associated factors in the northeastern Uganda region of Karamoja.

## Materials and methods

### Study design

This was a retrospective study which involved quantitative method of data collection and analysis. Using a data abstraction tool we abstracted relevant data from health facility TB treatment register. We included data for individuals notified with drug-susceptible tuberculosis and started on TB treatment in the period 1st January 2020 to 31st December 2021. Records that had treatment outcome were considered. The study data was collected in February 2023.

### Study setting

The study was conducted in St. Kizito Hospital Matany, Napak district, southern Karamoja. This is a Private Not-For-Profit (PNFP) hospital under the Catholic Diocese of Moroto (North-Eastern Uganda). The hospital capacity constitutes 226 beds and provides a comprehensive inpatient and outpatient medical services to the people of Karamoja and the neighboring regions. The Hospital is being supported by The AIDS Support Organization (TASO), funded by the U.S President's Emergency Plan for AIDS Relief (PEPFAR) through Center for Disease Control and Prevention (CDC) health system strengthening grant to implement high quality TB/HIV services. Karamoja region has an estimated population of 1.1million, literacy rate of 11% [19], HIV/AIDS prevalence rate of 3.9% [20], The region is characterized by limited access to health services, inadequate health infrastructure, insufficient quality of care, inadequate human and financial resources for health, and inadequate social protection [19, 21] and these underlying social, health, and demographic determinants need to be adequately addressed to achieve End TB global target. TB diagnosis in this facility is done using laboratory tests such as Gene Xpert, microscopy and urine TB lipoaribomanan (TB LAM) or through clinical assessment. Treatment for drug susceptible TB in this facility is done using standard world health organization (WHO) and national regiments [22, 23], that is, a combination of Rifampicin (R), Isoniazid (H), pyrazinamide (Z) and ethambutol (E) for 2 months and RH for 4 months.

### Inclusion and exclusion criteria

All individuals notified as new and relapse drug susceptible TB and started on TB treatment at the study site were eligible for inclusion in the study, while individuals with incomplete data on age, gender, and disease type and with no treatment outcome (not evaluated) and those with multi-drug resistant TB were excluded.

### Study variables

**Exposures.** Demographic characteristics (age, gender), residence (Napak or outside), disease characteristics (disease category, treatment type), disease classification categorized as bacteriologically confirmed (laboratory diagnosis with urine LAM, Gen Xpert or Miscroscopy), clinically diagnosed (based on clinical assessment and chest x-ray diagnosis) and extrapulmonary (a diagnosis of TB outside the lungs) [23], TB Patient type categorized as new or

retreatment, HIV status (Positive or negative), point of referral (Facility or community) and Nutritional status measured using mid upper arm circumference (MUAC). The MUAC codes documented in the register were captured and used to classify nutritional status of the study participants, that is, red (severe acute malunitrition), yellow (moderate acute malnutrition), and green (well nourished). The participants whose MUAC were not taken were categorized as unknown nutritional status.

**The primary outcome** was TB treatment Success defined as treatment completion and/or Cure. The primary outcome was measured as the proportion of drug susceptible TB patients started on treatment who at the end of 12 months was document in the register as completed treatment, and/or cured of TB. Unsuccessful treatment outcome included, Death, Lost-to-follow-up or treatment failure.

Definition of Treatment Outcomes

1. Successful outcome is when a TB patients is cured (negative smear microscopy at the end of treatment and on at least one previous follow-up test) or completed (A TB patient who completed treatment without evidence of failure but with no record to show that sputum smear or culture results in the last month of treatment and on at least one previous occasion were negative, either because tests were not done or because results are unavailable) [24].

2. Unsuccessful outcome is when treatment resulted in treatment failure (remaining smear-positive after 5 months of treatment), defaulted (patients who interrupted their treatment for two consecutive months or more after registration), died, Lost to follow-up (A TB patient who did not start treatment or who completed more than 1 month of treatment and was interrupted for 2 consecutive months or more) [24].

## Data management and analysis

Using a customized data abstraction tool, data was extracted on both the exposure and outcome variables from the health unit TB registers from the time of registration at the TB treatment until treatment outcome was reached. The collected data was entered into an electronic dataset using MS Excel and then exported to Stata version 15.0 where it was cleaned, validated, and analyzed. We summarized the study participant characteristics and TB treatment outcomes using descriptive statistics. We used Chi-square to test the difference in treatment success across the 2-year period. Modified poison regression analysis was conducted to determine factors associated with treatment success rate and reported as prevalence ratios of treatment success. While using modified Poisson regression, the standard errors were estimated using robust method, by applying variance-covariance matrix of the estimators (vce) method. Modified Poisson regression was chosen over logistic regression because, logistic regression overestimates relative risks and prevalence ratios in instances where the binary outcome has a high or moderate incidence/prevalence in the studied population [25]. First, we performed a bivariate analysis by regressing each independent variable to the outcome, and later subjected all variables with biological relevance or confounders and those with a P-value less than 0.1 to a multiple regression. The level of significance was set at 5%.

**Ethical approval and consent to participate.**   Ethical approval and informed consent waiver for the study was sought from TASO Uganda Research and Ethics Committee (REC), approval number TASO-2022-99. Confidentiality was maintained by ensuring that individual patient-level data obtained was de-identified, encrypted, and passworded to ensure access by only an authorized team of investigators. Administrative clearance to collect and utilize medical records of the hospital was sought from the Medical Superintendent- St. Kizito Hospital Matany.

## Results

From the table above, the median age of the 1234 participants was 31(IQR: 13–49)years, and 26.2%(n = 323) were children below 15years of age. About a half of the participants, 51.4% (n = 634) were bacteriologically confirmed with TB, while 22.4%(n = 277) were extrapulmonary TB cases. Ten percent (n = 130) of the participants were HIV/TB co-infected (Table 1).

### TB treatment outcomes

The treatment outcomes reported were; treatment completion at 52.4% (n = 647), cured at 26.9% (n = 332), death 10.4%(n = 129), Lost to follow up 9.8% (n = 121) and failed at 0.4% (n = 5).

### Trends in treatment success

From Fig 1 below, TB TSR improved from 77.9% in Jan-Mar 2020 to 87.7 in Oct-Dec 2021, with improvement more marked in the year 2021. The overall treatment success rate for the 2

**Table 1.  Characteristics of study participants.**

| Variable | Population, N = 1234 | Proportions (%) |
|---|---|---|
| Age/years. median(IQR) | 31(13–49) | |
| Body weight/Kg, Median (IQR) | 45(24–52) | |
| **Age Category** | | |
| <15years(Children) | 323 | 26.2 |
| 15_49years(Adults) | 610 | 49.4 |
| >49(Elderly) | 301 | 24.4 |
| **Sex** | | |
| Female | 541 | 43.8 |
| **Residence in Napak** | | |
| Yes | 845 | 68.5 |
| **Disease classification** | | |
| Bacteriologically confirmed | 634 | 51.4 |
| Clinically Diagnosed | 323 | 26.2 |
| Extrapulmonary | 277 | 22.4 |
| **Nutritional (MUAC)** | | |
| Green (Well nourished) | 366 | 29.7 |
| Yellow (Moderate undernourished) | 293 | 23.7 |
| Red (severe undernourished) | 322 | 26.1 |
| Unknown | 253 | 20.5 |
| **Treatment Model** | | |
| Facility based | 1 | 0.1 |
| Digital Community DOT | 4 | 0.3 |
| Non Digital comm DOT | 1229 | 99.6 |
| **Patient Type** | | |
| New | 1159 | 93.9 |
| Retreatment | 75 | 6.1 |
| **HIV status** | | |
| Negative | 1104 | 89.5 |
| Positive | 130 | 10.5 |
| **Year of treatment (Cohort)** | | |
| year 2020 | 560 | 45.4 |
| year 2021 | 674 | 54.6 |

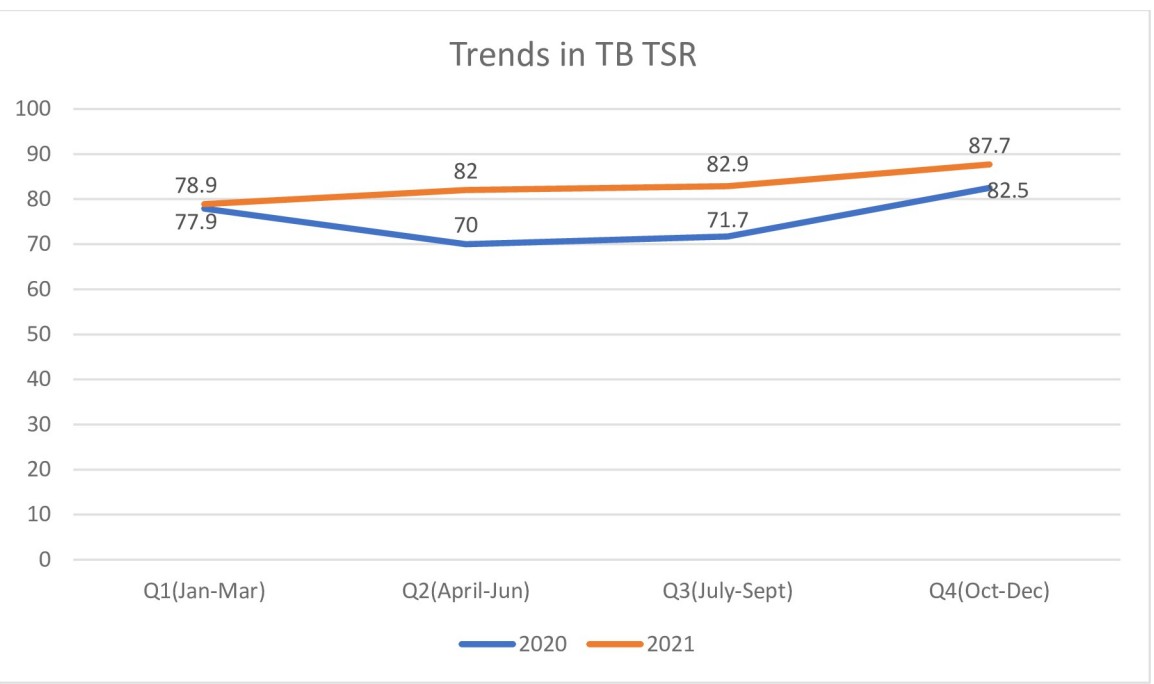

**Fig 1. Trends in TB treatment success rate for 2020 and 2021.**

years combined was 79.3%(95%CI; 77.0%-81.5%), while treatment success rate for 2020 was 75.4% and 2021 was 82.4% and this difference was sIginificant (P = 0.002).

From the Table 2 above, the factors that were negatively associated with treatment success were; Older age, under nutrition/unknown nutritional status (MUAC measurement red or unknown), and HIV positive status, that is.

Prevalence of treatment success was lower in adults (15–49 years) compared to children <15years, PR = 0.93(95%CI; 0.86–0.99, P = 0.036). The prevalence was also lower in the elderly patients > 49years compared to children, PR = 0.87(95%CI; 0.80–0.94, P = 0.001). Nutritional status. Those whose MUAC were red, PR = 0.91(95%CI; 0.85–0.98, P = 0.013) and those whose MUAC were unknown(not taken), PR = 0.80(95%CI; 0.73–0.88, P<0.0001) had lower a prevalence of treatment success as compared to clients whose MUAC were normal(green). HIV status. Patients who were HIV/TB coinfected had lower prevalence of treatment success, PR = 0.88(95%CI; 0.78–0.98, P = 0.026) as compared to clients who were HIV negative.

Year of enrollment was a positive predictor of treatment success. Patients who were registered/enrolled in 2021 had high prevalence of treatment success, PR = 1.09(95%CI; 1.03–1.16, P = 0.003).

## Discussion

This study was carried out to investigate TB drug-susceptible treatment success rate (DS-TB TSR) and its associated factors among individuals diagnosed and treated for TB between January 2020 through December 2021 in a private not-for-profit hospital in Karamoja, Uganda. The study was unique in that it investigated treatment success in a nomadic community which is highly mobile. In the study, we found that about a half (56%) of the participants completed TB treatment. However, poor TB treatment outcomes (such as lost-to-follow-up, TB relapse and death) were more common among HIV co-infected individuals, older individuals and

**Table 2. Factors associated with drug susceptibel tuberculosis treatment success rate (TSR) in Matany Hospital.**

| Variable | Overall, N = 1234 | Treatment Success, n(%),n = 979 | Treatment Failure, n(%), n = 255 | cPR(95%CI) | aPR(95%CI) | P-value |
|---|---|---|---|---|---|---|
| **Age Category** | | | | | | |
| <15years(Children) | 323 | 271(83.9) | 52(16.1) | 1 | 1 | |
| 15_49years(Adults) | 610 | 485(79.5) | 125(20.5) | **0.95(0.89–1.01)** | **0.93(0.86–0.99)** | **0.036** |
| >49(Elderly) | 301 | 223(74.1) | 78(25.9) | **0.88(0.81–0.96)** | **0.87(0.80–0.94)** | **0.001** |
| **Sex** | | | | | | |
| Female | 541 | 432(79.9) | 109(20.1) | 1 | 1 | |
| Male | 693 | 547(78.9) | 146(21.1) | 0.99(0.93–1.05) | 0.99(0.93–1.04) | 0.655 |
| **Residence in Napak** | | | | | | |
| No | 389 | 305(78.4) | 84(21.6) | 1 | 1 | |
| Yes | 845 | 674(79.8) | 171(20.2) | 1.02(0.96–1.08) | 1.00(0.94–1.06) | 0.954 |
| **Disease classification** | | | | | | |
| Bacteriologically confirmed | 634 | 512(80.8) | 122(19.2) | 1 | 1 | |
| Clinically Diagnosed | 323 | 253(78.3) | 70(26.7) | 0.97(0.91–1.04) | 0.95(0.88–1.03) | 0.232 |
| Extrapulmonary | 277 | 214(77.3) | 63(22.7) | 0.96(0.89–1.03) | 0.93(0.87–1.01) | 0.075 |
| **Nutritional (MUAC)** | | | | | | |
| Green | 366 | 310(84.7) | 56(15.3) | 1 | 1 | |
| Yellow | 293 | 243(82.9) | 50(17.1) | 0.98(0.91–1.05) | 0.99(0.92–1.06) | 0.709 |
| Red | 322 | 254(78.9) | 68(21.1) | **0.93(0.87–1.00)** | **0.91(0.85–0.98)** | **0.013** |
| Unknown | 253 | 172(68.0) | 81(32.0) | **0.80(0.73–0.88)** | **0.80(0.73–0.88)** | **0.000** |
| **Patient Type** | | | | | | |
| New | 1159 | 925(79.8) | 234(20.2) | 1 | 1 | |
| Retreatment | 75 | 54(72.0) | 21(28.0) | 0.90(0.78–1.04) | 0.91(0.79–1.04) | 0.174 |
| **HIV status** | | | | | | |
| Negative | 1104 | 889(80.5) | 215(19.5) | 1 | 1 | |
| Positive | 130 | 90(69.2) | 40(30.8) | **0.86(0.76–0.97)** | **0.88(0.78–0.98)** | **0.026** |
| **Year of treatmemt (Cohort)** | | | | | | |
| Year 2020 | 560 | 422(75.4) | 138(24.6) | 1 | 1 | |
| Year 2021 | 674 | 557(82.6) | 117(17.4) | **1.10(1.03–1.16)** | **1.09(1.03–1.16)** | **0.003** |

NB: **Bold** = significant with P<0.05, cPR = crude prevalence ratio, aPR = adjusted prevalence ratio.MUAC = Mid upper arm circumference All factors were adjusted for each other, Treatment success = cured and/or completed, Treatment failure = died, lost to follow up or failed.

those with severe malnutrition. The prevalence of TB treatment success was higher in 2021 compared to 2020. Now there is new data indicating that the ramifications of COVID-19 on health and health systems affected the delivery of health care, especially in 2020 [26]. We think that the lower TSR in 2020 were as a result of these effects of COVID-19.

Overall, eight in ten individuals with drug-sensitive TB achieved treatment success which compares well with the national TSR [14], but both are sub-optimal compared to the WHO End TB strategy TSR target of nine in ten [2]. Our findings show a significant improvement from earlier estimates of five in ten treatment success in the region [16], suggesting that there has been improvement in the delivery of TB care services. These acheivements could be as a result of the recent governemnt deliberate efforts to break the transmission cycle for TB through early diagnosis and treatment of both cases and their contacts. Similar to our findings, treatment success rates of 63.9–82.1% [12, 27] were also observed from studies conducted in other rural regions of the country. In SSA sub-optimal TSR is not unique to Uganda alone as a systematic review summarizing TSR among adult bacteriologically confirmed pulmonary tuberculosis patients in the sub continent showed a similar trend [4]. These findings are

pointers towards a need for novel, innovative and evidence based interventions designed towards improvement of performance of the national TB program.

In our study, the TSR improved from 75.4% in 2020 to 82.4% in 2021, which is a progressive improvement better than a stagnation or a reversal. Besides the receovery of health services post COVID-19 lockdowns, we also attribute this performance to Continuous Quality Intervention (CQI) projects by The AIDS Support Organization (TASO) and other implementing partners introduced to address the low TSR in the region. Specifically, the TSR retention strategies which entailed introduction and use of an appointment system in the TB Clinic, timely follow up of missed appointments and active cohort monitoring monthly were introduced.

In the current study, older individuals, compared to younger ones, had a lower prevalence of treatment success. This is comparable to what has been reported in other studies such as that by, Izudi et al., who reported that successful treatment of TB was less likely to occur among patients older than 50 years [12]. Similarly, Sebuliba et al found that older individuals (aged 65 + years) had two-fold increased odds of poor TB treatment outcomes, compared to younger individuals, in a Kampala cohort [28]. Older individuals are more likely to have co-morbidities such as hypertension that increase chances of poor TB treatment outcomes [29]. Moreover, older individuals are less likely to have treatment supporter and more likely to suffer from anti-TB drug toxicity such as hepatotoxicity and renal toxicity that also increase the chances of poor treatment outcomes [30, 31]. It is therefore not surprising that patients aged > 49 years had a lower prevalence of treatment success compared to the younger patients. This highlights the need to closely monitor and follow up elderly patients on anti-TB chemotherapy.

Furthermore, we found that HIV positive TB patients had a lower prevalence of TSR than HIV negative individuals. This is similar to findings reported in studies done in other high burden TB countries in Sub-Saharan Africa [32, 33], America [34, 35] and Asia [10, 11]. HIV positive patients had a lower prevalence of treatment success probably due to the drug-drug interactions between anti-TB chemotherapy and the highly active antiretroviral therapy (HAART) and a high pill burden that may affect the adherence to the anti-TB chemotherapy [36, 37]. Non-adherence to anti-TB chemotherapy can in turn lead to a low TSR. Strengthening of HIV and TB treatment and public health programs is thus needed to improve the TSR among patients with HIV.

Under nutrition is a well-established factor associated with a low TB TSR [38]. Our study also collaborates this finding. Integrating nutritional support into TB treatment programs is thus key in improving TB treatment success rates.

Our study had some limitations, for example, data was collected for a period of 2years, which was not sufficient to demonstrate long term trends in TSR. The use of retrospective data presented some levels of data incompleteness. The treatment outcome was documented in the register only at 12 months after initiation of treatment and not at the point of event. Being a retrospective study, we were unable to obtain some key information which could have influenced our study outcomes. The characteristics such as mobility and accessibility, cultural practices, livelihood priorities, socioeconomic factors and education, were not assessed yet they may impact TB treatment outcomes among pastoralist communities.

Despite these challenges, data were collected from a private-not-for-profit tertiary care facility, and it could be one of the first studies investigating TB TSR in such a setting. This could imply and depict the real practice in many private-not-for-profit tertiary hospitals in Uganda. Also, the sample size was large and thus results could be generalized to other similar settings.

## Conclusion

TB Treatment success rate (TSR) in Matany hospital was suboptimal. Older age, poor nutrition and being HIV-positive were negative predictors of treatment success. We propose integrating

nutrition and HIV care into TB programming to improve treatment success. Additional research on understanding of TB treatment challenges and enablers among the elderly is important to design tailored interventions for this group. Additional research is needed to determine at what point most clients fail treatment, get lost to follow up or die after initiating TB treatment so as to inform evidence-based guidelines on optimization TSR.

## Supporting information

**S1 Data.**
(CSV)

## Acknowledgments

The authors would like to acknowledge the support rendered by the Medical Superintendent of St.Kisito Hospital-Matany, Dr.Nsubuga JB during the process of data collection for this study. We would also like to acknowledge the contribution of The AIDS Support Organization (TASO) in supporting TB program implementation in Matany Hospital.

## Author Contributions

**Conceptualization:** Ronald Opito, Keneth Kwenya, Saadick Mugerwa Ssentongo, Mark Kizito, Susan Alwedo, Baker Bakashaba, Yunus Miya, Lameck Bukenya, Eddy Okwir, Lilian Angwech Onega, Andrew Kazibwe, Fred Kirya.

**Data curation:** Ronald Opito, Saadick Mugerwa Ssentongo, Baker Bakashaba, Yunus Miya, Lameck Bukenya, Eddy Okwir, Lilian Angwech Onega, Andrew Kazibwe, Emmanuel Othieno, Fred Kirya, Peter Olupot Olupot.

**Formal analysis:** Ronald Opito, Keneth Kwenya, Mark Kizito, Susan Alwedo, Yunus Miya, Lilian Angwech Onega, Peter Olupot Olupot.

**Funding acquisition:** Ronald Opito.

**Investigation:** Ronald Opito.

**Methodology:** Ronald Opito, Keneth Kwenya, Saadick Mugerwa Ssentongo, Mark Kizito, Baker Bakashaba, Lameck Bukenya, Eddy Okwir, Andrew Kazibwe, Emmanuel Othieno, Fred Kirya, Peter Olupot Olupot.

**Project administration:** Ronald Opito, Saadick Mugerwa Ssentongo.

**Resources:** Ronald Opito.

**Supervision:** Peter Olupot Olupot.

**Validation:** Ronald Opito, Peter Olupot Olupot.

**Visualization:** Peter Olupot Olupot.

**Writing – original draft:** Ronald Opito, Keneth Kwenya, Saadick Mugerwa Ssentongo, Mark Kizito, Susan Alwedo, Baker Bakashaba, Yunus Miya, Lameck Bukenya, Eddy Okwir, Lilian Angwech Onega, Andrew Kazibwe, Emmanuel Othieno, Fred Kirya.

**Writing – review & editing:** Ronald Opito, Saadick Mugerwa Ssentongo, Peter Olupot Olupot.

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
