## [Decision Letter · Decision Letter 0]

24 Jan 2024

PONE-D-23-35125Tuberculosis treatment success rate, associated factors and outcomes in St.Kizito Hospital, Matany, Napak district, Karamoja region. A retrospective study.PLOS ONE

Dear Dr. MUGERWA SSENTONGO,

Thank you for submitting your manuscript to PLOS ONE. After careful consideration, we feel that it has merit but does not fully meet PLOS ONE’s publication criteria as it currently stands. Therefore, we invite you to submit a revised version of the manuscript that addresses the points raised during the review process.

We look forward to receiving your revised manuscript.

Kind regards,

Hamufare Dumisani Dumisani Mugauri, Ph.D. Public Health

Academic Editor

PLOS ONE

Journal Requirements:

This publication was partially (data collection) supported through SUNRIF, Soroti University research and innovation fund, Round1, Award No SUNRIF 2022/22 to Ronald Opito. The views and opinions of the authors expressed herein do not necessarily state or reflect those of the funder.

4. Please amend the manuscript submission data (via Edit Submission) to include author Dr. Ronald Opito.

Reviewers' comments:

Reviewer's Responses to Questions

**Comments to the Author**

1. Is the manuscript technically sound, and do the data support the conclusions?

Reviewer #1: Partly

Reviewer #2: Yes

Reviewer #3: Partly

2. Has the statistical analysis been performed appropriately and rigorously? 

Reviewer #1: Yes

Reviewer #2: Yes

Reviewer #3: No

3. Have the authors made all data underlying the findings in their manuscript fully available?

Reviewer #1: No

Reviewer #2: Yes

Reviewer #3: Yes

4. Is the manuscript presented in an intelligible fashion and written in standard English?

Reviewer #1: No

Reviewer #2: Yes

Reviewer #3: Yes

5. Review Comments to the Author

Reviewer #1: Thanks for submitting the manuscript to PLOS one

Interesting data about the clinical outcomes in TB patients in Napak District, Southern Karamoja, However I have several concerns and comments below.

Title: I suggest author to be clearly about what is the outcome, author should state for example associated factors and treatment success. and this study includes only the TB drug susceptible so author should state the population clearly in the Title.

- Journal: Please check again for the journal Format and should follow journal format.

-Language: Please check and proof the English Language again throughout the whole manuscript.

- All abbreviations should state the full name first.

Introduction

- Line 54: I don't quite understand "single infectious agent" please use other word.

- Please add more previous study about factors associated with treatment success or failure and tell more about your study gap of knowledge why author would like to conduct this study. Are there any differences about the factors between your population VS others

Methods

- What is customized proforma, may be used other word.

-Study setting. Because we have more than one method for TB diagnosis and also TB laboratories. Author have to state how your institutes register patients with TB, what is your criteria for TB diagnosis and LAB (PCR, AFB or others)

Inclusion: Why author includes only TB drug susceptible patients?

Definition: To make reader understand more, Authors have to give the definition >> New and relapse drug susceptible TB by the standard and reliable guideline. Author have to give the definition for all the treatment outcomes and which guideline author used for the reference.

Statistical analysis: This study is retrospective study. I would like to ask author why author use poison rather than logistic regression for factors associated with treatment outcome.

Result.

Table 1 should add more data: Body weight, underlying disease. For sex and residence can show only female or yes.

For table 1 please add the definition of disease classification in the method session.

For table 1 add more data about the TB drug regimens and duration of anti-TB drug treatment

For table 1 add more data about patients' compliance, % Hospitalization, IPD or OPD patients.

Table2: add the full name for all the abbreviation at the footnote under the table ex. cPR,aPR

For Table 2 I suggest author the show only one outcome (Treatment success or failure).

Discussion

I don't quite understand the phase ".......... in a nomadic community which is highly mobile". please clarify

" In the study, we found that a majority of the participants complete TB treatment" In my opinion, 56% of patients are not the majority.

Please discuss the prevalence in your study VS others

Please check the references, some paraphrase didn't state the Ref. ex. This is comparable to what has been in other studies (Ref.???).

-Please clarify more why Hypertension and endocrine abnormalities associated with poor outcome ?

-Please clarify more why Drug interaction related to HIV positive and low prevalence of treatment failure

- What is your strength and limitation

- I suggest author to add the discussion more about the system between the year 2020 and 2021. Are there any different systems between these 2 years because this study's result showed the different treatment outcome.

Reference

Check all the format of the references

Hope these comments might help authors for better manuscript.

Reviewer #2: The authors report a retrospective study on the Tuberculosis treatment success rate, associated factors and outcomes in St.Kizito Hospital, Matany, Napak district, Karamoja region. I have several concerns that should be addressed

General

The authors need to proofread their manuscript to correct all grammatical errors in the manuscript. I have identified several problems with grammar/spelling/punctuation. For example, the word “repported” in line 44 is misspelt, while writing 1st January 2020 in the Study Design paragraph, the “st” should be Superscripted, to mention a few.

Introduction

The authors started using the term DR-NCDs without first explaining the meaning of it. I know this has been explained in the abstract, but it is recommended that when you start writing the main text, all the initials are explained again. The authors should make sure all the abbreviations are explained before starting to use them.

Results

Generally, the authors should not use abbreviations in the titles of the tables. The rule is also for a table to be able to stand alone and be understandable, so it is recommended that all abbreviations used in the table should be explained below the table, as footnotes.

In Table 1, Nutritional (MUAC) is classified as Green, Yellow, and Red without clarifying what the colours represent.

I also suggest that the results for Table 2 should be re-written because the way the paragraph is written now is confusing. My suggestion is that the authors could first mention all the factors and then start explaining them one by one.

Discussion

While reporting the limitation of the study, the authors should have mentioned the inability to obtain data which might have influenced their results. The characteristics that may impact TB treatment outcomes among pastoralist communities include Mobility and Accessibility, Cultural Practices, Livelihood Priorities, Socioeconomic Factors, Education Levels etc.

Reviewer #3: 1. Inclusion and exclusion: I am afraid that excluding those with missing age, gender and disease type would introduce bias in the estimates. The authors need to assess if complete case analysis did not produce biased results. By doing multiple imputation and then compare results (complete case vs imputed data) to check if there is consistence in the findings would help.

2. Inclusion and exclusion: Since those not evaluated were also initiated on treatment, I think they should be part of the denominator. Excluding them especially if they are many can lead to overestimation of the treatment success rate. Can the authors estimate the treatment success rate with the "not evaluated" included?

3. Inclusion and exclusion: Please clarify why the multi-drug resistant TB patients were excluded.

4. Under methods and materials, the authors mention to have used a customized proforma to abstract data. A reference to this proforma at its first mention would be helpful.

5. Study variables: A clear definition of the primary outcome should be given. I suppose this was a proportion. If yes, please define the primary outcome mentioning the numerator and the denominator.

6. Under data management and analysis, the authors mention to have used a modified poison regression to estimate prevalence ratios. Did the authors fit the model with robust standard errors? Since Poison regression if used for binary outcome, it can estimate confidence interval for the prevalence that exceeds a 1, and consequently this affects estimates for the prevalence ratio. Please check that robust standard errors were used and mention this in the analysis methods.

7. It's not clear how a multivariable Poison regression model was built. What factors were considered in the multivariable model and how did the authors arrive at a final fit. Also, how did they decide which factors to include in the model? All these details are needed to make things clear and for reproducibility reasons.

8. Under results in Table 2, it seems like all variables in the data were included in the multivariable analysis and no model building was considered. Would the results be different if only variables with say a p-value at bivariable analysis less than 0.1 were included? Did the authors check for multicollinearity? What about overdispersion, noting that Poisson model assumes equality of variance and mean, which assumption if it does not hold, the model can give misleading results. A negative binomial would be helpful.

9. In reference to my point 1, would the results in Table 2 remain the same if missing data were considered (instead of excluding them)?

10. Below table 2, statements like "Nutrition status", "HIV status" and others look like hanging statements. Please revise the paragraph as you list the significant factors.

11. Please add line numbers to the manuscript as this will make the review process easier.

12. Somewhere you are missing full stops. For example, a full stop is missing before the statement that begins "We used chis-square to test ..."

6. PLOS authors have the option to publish the peer review history of their article (what does this mean?). If published, this will include your full peer review and any attached files.

Reviewer #1: No

Reviewer #2: No

Reviewer #3: No

---

## [Author Response · Author response to Decision Letter 0]

20 Feb 2024

The Editor 

PlosOne Journal 

Dear Professor, Hamufare Dumisani Dumisani Mugauri,

REF: RESPONSE TO THE ACADEMIC EDITOR AND REVIEWERS TO THE ARTICLE: 

Title: “Tuberculosis treatment success rates, associated factors and outcomes in St.Kizito Hospital, Napak district, Karamoja region, A retrospective study” 

I on behalf of the authors would like to express our sincere gratuity for the positive consideration you have given this article and for the positive feedback so far given by the academic editor and reviewers. 

Please find highlighted in red our responses to reviewers and academic editor’s comments. 

Reviewer #1: Thanks for submitting the manuscript to PLOS one

Interesting data about the clinical outcomes in TB patients in Napak District, Southern Karamoja, However I have several concerns and comments below.

Title: I suggest author to be clearly about what is the outcome, author should state for example associated factors and treatment success. and this study includes only the TB drug susceptible so author should state the population clearly in the Title.

Response: The title has been modified as per the recommendation to, “Treatment success rate and associated factors among drug susceptible tuberculosis individuals in St.Kizito Hospital, Matany, Napak district, Karamoja region. A retrospective study.” 

- Journal: Please check again for the journal Format and should follow journal format.

Response: This has been checked and the manuscript edited in line with the journal format.

-Language: Please check and proof the English Language again throughout the whole manuscript.

Response: This has been checked and proofreading done. 

- All abbreviations should state the full name first.

Response: All abbreviations have been explained at their first mention. 

Introduction

- Line 54: I don't quite understand "single infectious agent" please use other word.

Response. This has been modified as indicated in line 59.

- Please add more previous study about factors associated with treatment success or failure and tell more about your study gap of knowledge why author would like to conduct this study. Are there any differences about the factors between your population VS others

Response. This has been addressed in the introduction, lines 74-76. 

Methods

- What is customized proforma, may be used other word.

Response: We have changed to data abstraction tool. 

-Study setting. Because we have more than one method for TB diagnosis and TB laboratories. Authors have to state how your institutes register patients with TB, what is your criteria for TB diagnosis and LAB (PCR, AFB or others)

Response. This has been added, line 112-117. 

Inclusion: Why author includes only TB drug susceptible patients?

In Uganda (and more specifically Matany Hospital), Multi drug-resistant TB (MDR-TB) patients receive extra support such as food, and transport among others. These enablers have a high potential of biasing the outcome which will eventually give good treatment success rates. We deliberately left out MDR-TB patients to allow us to eliminate this potential bias. 

Definition: To make reader understand more, Authors have to give the definition >> New and relapse drug susceptible TB by the standard and reliable guideline. Author have to give the definition for all the treatment outcomes and which guideline author used for the reference.

Response: This has been well defined, lines 154-166. 

Statistical analysis: This study is retrospective study. I would like to ask author why author use poison rather than logistic regression for factors associated with treatment outcome.

Response: We applied modified Poisson with robust error estimations (rather than just poisson) because logistic regression overestimates relative risks/prevalence ratios more so if the binary outcome is high or moderate, more than 10% (Gnardelis C et al 2022; Misuse of logistic modelling). 

Result.

Table 1 should add more data: Body weight, underlying disease. For sex and residence can show only female or yes.

Response. Body weight has been added to table1. Other underlying diseases other than HIV are not routinely captured on the TB register. The sex and residence presentations have been adjusted as advised.

For table 1 please add the definition of disease classification in the method session.

Response: This has been added, lines 125-128. 

For table 1 add more data about the TB drug regimens and duration of anti-TB drug treatment

Response: Treatment for drug susceptible TB in this facility is done using standard national regiments, that is, for a combination of Rifampicin (R), Isoniazid (H), pyrazinamide (Z) and ethambutol (E) for 2 months and RH for 4 months.

So we felt it was unnecessary to add since it does not vary from patient to patient. We have however added this information on the methods section, lines 114-117. 

For table 1 add more data about patients' compliance, % Hospitalization, IPD or OPD patients.

Response: We collected data from the TB register which does not specify whether the clients were admitted however most of the clients are managed as OPD and the adherence is not routinely documented in the TB treatment register.

Table2: add the full name for all the abbreviation at the footnote under the table ex. cPR, aPR.

Response. This has been added at the footnote. 

For Table 2 I suggest author the show only one outcome (Treatment success or failure).

Response: We appreciate the suggestion to show only one outcome (treatment success for our case). However, we feel it causes no harm to have both outcomes shown, for purposes of clarity. 

Discussion

I don't quite understand the phase ".......... in a nomadic community which is highly mobile". please clarify

Response. The study population come from a pastoralist community, and they are quite mobile as they move from one place to another in search of pasture and water for their animals. The settlement arrangements are temporary. This influences treatment outcomes, more so on the lost to follow up. 

" In the study, we found that a majority of the participants complete TB treatment" In my opinion, 56% of patients are not the majority.

Response. We acknowledge this recommendation and agree with the reviewer. We have changed the statement. 

Please discuss the prevalence in your study VS others

Please check the references, some paraphrase didn't state the Ref. ex. This is comparable to what has been in other studies (Ref.???).

Response: This has been discussed, the references have been inserted. 

-Please clarify more why Hypertension and endocrine abnormalities associated with poor outcome?

Response: A reference on the interactions between hypertension and TB TSR has been included. Line 273. 

-Please clarify more why Drug interaction related to HIV positive and low prevalence of treatment failure

Response: “HIV positive patients had a lower prevalence of treatment success probably due to the drug-drug interactions between anti-TB chemotherapy and the highly active antiretroviral therapy (HAART) and a high pill burden that may affect the adherence to the anti-TB chemotherapy.” This is what we have indicated in our discussion section, which is well referenced. HIV positivity is associated with low prevalence of treatment success, and not the other way round. 

- What is your strength and limitation

Response: The strengths and limitations of the study have been presented as part of the discussions, lines 293-306. 

- I suggest author to add the discussion more about the system between the year 2020 and 2021. Are there any different systems between these 2 years because this study's result showed the different treatment outcome.

Response: The systems were not different as per say, however, there were programmatic interventions done by The AIDS Support Organization (TASO) geared towards improving TB treatment outcomes in Karamoja region. These interventions included quality improvement projects at facility and community levels. These have been discussed in the discussion sections, lines 228-265. 

Reference

Check all the format of the references.

Response: This has been checked and corrected. 

Hope these comments might help authors for better manuscript.

Reviewer #2: The authors report a retrospective study on the Tuberculosis treatment success rate, associated factors and outcomes in St.Kizito Hospital, Matany, Napak district, Karamoja region. I have several concerns that should be addressed

General

The authors need to proofread their manuscript to correct all grammatical errors in the manuscript. I have identified several problems with grammar/spelling/punctuation. For example, the word “repported” in line 44 is misspelt, while writing 1st January 2020 in the Study Design paragraph, the “st” should be Superscripted, to mention a few.

Response: The manuscript has been proofread and the grammatical errors corrected. 

Introduction

The authors started using the term DR-NCDs without first explaining the meaning of it. I know this has been explained in the abstract, but it is recommended that when you start writing the main text, all the initials are explained again. The authors should make sure all the abbreviations are explained before starting to use them.

Response: All abbreviations have been explained and written in full at first mention. 

Results

Generally, the authors should not use abbreviations in the titles of the tables. The rule is also for a table to be able to stand alone and be understandable, so it is recommended that all abbreviations used in the table should be explained below the table, as footnotes.

Response. The recommendation has been taken into consideration and all abbreviations used in the tables explained below as footnotes. The title abbreviations have been explained equally in their first mention. 

In Table 1, Nutritional (MUAC) is classified as Green, Yellow, and Red without clarifying what the colours represent.

Response. The meaning of each color has been explained under methodology, lines 131-134. 

I also suggest that the results for Table 2 should be re-written because the way the paragraph is written now is confusing. My suggestion is that the authors could first mention all the factors and then start explaining them one by one.

Response. This suggestion is noted and adopted for clarity. 

Discussion

While reporting the limitation of the study, the authors should have mentioned the inability to obtain data which might have influenced their results. The characteristics that may impact TB treatment outcomes among pastoralist communities include Mobility and Accessibility, Cultural Practices, Livelihood Priorities, Socioeconomic Factors, Education Levels etc.

Response. This recommendation is much appreciated and has been included as part of the study limitations, lines 297-301. 

Reviewer #3: 1. Inclusion and exclusion: I am afraid that excluding those with missing age, gender and disease type would introduce bias in the estimates. The authors need to assess if complete case analysis did not produce biased results. By doing multiple imputation and then compare results (complete case vs imputed data) to check if there is consistence in the findings would help.

Response: The total number of excluded data was 35 (of which 10 were not evaluated and 25 had missing values). We think this proportion, being less than 5% is small compared to the overall and may not have had significant effect on the outcome. The effect if any is countered by the large sample size we used. 

2. Inclusion and exclusion: Since those not evaluated were also initiated on treatment, I think they should be part of the denominator. Excluding them especially if they are many can lead to overestimation of the treatment success rate. Can the authors estimate the treatment success rate with the "not evaluated" included?

Response: We found only 10 participants were not evaluated. We think this number is too small to affect the overall outcome, especially that our sample size was quite large, and as such we decided to exclude them from the analysis. 

3. Inclusion and exclusion: Please clarify why the multi-drug resistant TB patients were excluded.

In Uganda Multi drug-resistant TB patients receive extra support such as food, and transport among others. These enablers have a high potential of biasing the outcome which will eventually give good treatment success rates.

4. Under methods and materials, the authors mention to have used a customized proforma to abstract data. A reference to this proforma at its first mention would be helpful.

Response. The customized proforma was data abstraction tool designed based on the durg susceptible TB treatment register. The name has been changed to data abstraction tool as suggested by reviewer1. Lines 87

5. Study variables: A clear definition of the primary outcome should be given. I suppose this was a proportion. If yes, please define the primary outcome mentioning the numerator and the denominator.

Response: The primary outcome is well defined, lines 154-166. 

6. Under data management and analysis, the authors mention to have used a modified poison regression to estimate prevalence ratios. Did the authors fit the model with robust standard errors? Since Poison regression if used for binary outcome, it can estimate confidence interval for the prevalence that exceeds a 1, and consequently this affects estimates for the prevalence ratio. Please check that robust standard errors were used and mention this in the analysis methods.

Response. Yes, we used robust standard errors to fit the model. This has now been fully explained under the data analysis section, lines 177-185. 

7. It's not clear how a multivariable Poison regression model was built. What factors were considered in the multivariable model and how did the authors arrive at a final fit. Also, how did they decide which factors to include in the model? All these details are needed to make things clear and for reproducibility reasons.

Response. The details of how the models were built is now well explained under the data analysis section, lines 182-185. 

8. Under results in Table 2, it seems like all variables in the data were included in the multivariable analysis and no model building was considered. Would the results be different if only variables with say a p-value at bivariable analysis less than 0.1 were included? Did the authors check for multicollinearity? What about overdispersion, noting that Poisson model assumes equality of variance and mean, which assumption if it does not hold, the model can give misleading results. A negative binomial would be helpful.

Response: Multicollinearity and over-dispersion were checked and that’s how body weight and treatment models were dropped off from the final model. All the other variables included either had biological relevance, confounders or had a P-Value less than 0.1 at bivariate level. 

9. In reference to my point 1, would the results in Table 2 remain the same if missing data were considered (instead of excluding them)?

Response: The total number of excluded data was 35 (of which 10 were not evaluated and 25 had missing values). We think this proportion, being less than 5% is small compared to the overall and may not have had significant effect on the outcome. The effect if any is countered by the large sample size we used. 

10. Below table 2, statements like "Nutrition status", "HIV status" and others look like hanging statements. Please revise the paragraph as you list the significant factors.

Response. Revisions have been made as recommended. 

11. Please add line numbers to the manuscript as this will make the review process easier.

Response: The line numbers have been added. 

12. Somewhere you are missing full stops. For example, a full stop is missing before the statement that begins "We used chis-square to test ..."

Response: The revision has been made. 

Dr.Ssentongo Saadick

Corresponding Author

---

## [Decision Letter · Decision Letter 1]

7 Mar 2024

Treatment success rate and associated factors among drug susceptible tuberculosis individuals in St.Kizito Hospital, Matany, Napak district, Karamoja region. A retrospective study.

PONE-D-23-35125R1

Dear Dr. Sentongo,

We’re pleased to inform you that your manuscript has been judged scientifically suitable for publication and will be formally accepted for publication once it meets all outstanding technical requirements.

Within one week, you’ll receive an e-mail detailing the required amendments. When these have been addressed, you’ll receive a formal acceptance letter, and your manuscript will be scheduled for publication.

An invoice will be generated when your article is formally accepted. Please note, if your institution has a publishing partnership with PLOS and your article meets the relevant criteria, all or part of your publication costs will be covered. Please make sure your user information is up to date by logging into Editorial Manager at http://www.editorialmanager.com/pone/ and clicking the ‘Update My Information' link at the top of the page. If you have any questions relating to publication charges, please contact our Author Billing department directly at authorbilling@plos.org.

Kind regards,

Hamufare Dumisani Dumisani Mugauri, Ph.D. Public Health

Academic Editor

PLOS ONE

Reviewers' comments:

Reviewer's Responses to Questions

**Comments to the Author**

1. If the authors have adequately addressed your comments raised in a previous round of review and you feel that this manuscript is now acceptable for publication, you may indicate that here to bypass the “Comments to the Author” section, enter your conflict of interest statement in the “Confidential to Editor” section, and submit your "Accept" recommendation.

Reviewer #2: All comments have been addressed

Reviewer #3: All comments have been addressed

2. Is the manuscript technically sound, and do the data support the conclusions?

Reviewer #2: Yes

Reviewer #3: Yes

3. Has the statistical analysis been performed appropriately and rigorously? 

Reviewer #2: Yes

Reviewer #3: Yes

4. Have the authors made all data underlying the findings in their manuscript fully available?

Reviewer #2: Yes

Reviewer #3: Yes

5. Is the manuscript presented in an intelligible fashion and written in standard English?

Reviewer #2: Yes

Reviewer #3: No

6. Review Comments to the Author

Reviewer #2: The authors have addressed all of the comments which I raised previously. I have no further comments

Reviewer #3: All my comments were addressed. There are however a few typos, which I suggest to authors to pay keen attention or consider sending their paper to an English reviewer. Examples below;

1. In Line 164, "A TB" patient should be "a TB patient"

2. Line 185, "P-value" should be "p-value"

3. Line 208, "susceptibel" should be "susceptible"

4. Line 129, "Patient" should be "patient", "Positive" should be "positive"

5. Line 137, "was document" should be "was documented"

6. etc...

7. PLOS authors have the option to publish the peer review history of their article (what does this mean?). If published, this will include your full peer review and any attached files.

Reviewer #2: No

Reviewer #3: No

---

## [Editor Report · Acceptance letter]

12 Mar 2024

PONE-D-23-35125R1 

PLOS ONE

Dear Dr. Ssentongo, 

I'm pleased to inform you that your manuscript has been deemed suitable for publication in PLOS ONE. Congratulations! Your manuscript is now being handed over to our production team.

Kind regards, 

on behalf of

Mr Hamufare Dumisani Dumisani Mugauri 

Academic Editor

PLOS ONE